# Analysis on the Effects of External Temperature and Welding Speed on the Safety of EVA Waterproofing Sheet Joints by Hot Air Welding

**DOI:** 10.3390/ma13235586

**Published:** 2020-12-07

**Authors:** Wan-Goo Park, Su-Young Choi, Jin-Sang Park, Dong-Bum Kim, Xing-Yang He, Sang-Keun Oh

**Affiliations:** 1Architecture of Graduate School, Seoul National University of Science and Technology, 232 Gongneung-ro, Nowon-gu, Seoul 01811, Korea; dhdkdrn@naver.com (W.-G.P.); csyoung777@gmail.com (S.-Y.C.); 2New Material and Laboratory Co., Ltd., 17 Dasan Jungang-ro 19th Street, Namyangju-si 12248, Korea; sciencewater@naver.com (J.-S.P.); db2128@naver.com (D.-B.K.); 3School of Civil Engineering, Architecture and Environment, Hubei University of Technology, Wuhan 430068, China; hxycn@126.com; 4School of Architecture, Seoul National University of Science & Technology, 232 Gongneung-ro, Nowon-gu, Seoul 01811, Korea

**Keywords:** synthetic polymer, EVA waterproofing sheet, sheet joint, hot air welder, tensile performance, optimal welding speed

## Abstract

This study analyzes the optimal seasonal ambient temperature during welding and welding speed conditions for securing high tensile strength of ethylene vinyl acetate (EVA) waterproofing sheets bonded for roofing, installed by hot air welded joints (overlaps). Seven separate ambient temperature conditions (−10, −5, and 0 °C for winter conditions, 20 °C for the normal condition, and 25, 30, and 35 °C for summer conditions) were set for the test variable and seven speed conditions (3, 4, 5, 6, 7, 8, and 9 m/min) for hot air welding. Based on these conditions, EVA sheet joint specimens were prepared, and the tensile strength of the joint sections was tested and measured. Tensile strength results, compared to normal temperature conditions (20 °C) showed an increase in the summer temperature condition but a decrease during winter temperature conditions. The analysis on the effects of the welding speed showed that in summer temperature conditions (25, 30, and 35 °C), the optimum hot air welding speed is 4.3~9.0 m/min at 25 °C, 4.7~8.7 m/min at 30 °C and 5.2~8.6 m/min at 35 °C, whereas in winter (−10, −5, and 0 °C), the optimum hot air welding temperature is 3~4.1 m/min at −10 °C, 3~4.6 m/min at −5 °C and 3~4.9 m/min at 0 °C. Research results demonstrate that it is imperative to consider the welding speed in accordance to the respective seasonal temperature conditions to secure construction quality of the EVA joints for roofing.

## 1. Introduction

The waterproofing materials and methods applied in the construction sector are being developed in various forms as technology advances. However, the problem of roof leaks in buildings still persists. The contraction and expansion of roof slabs due to temperature and humidity are subject to physical environmental factors influencing cracks and joints [1]. These circumstances call for a requirement for waterproofing materials for roofs to retain high strength properties to respond to weather-related degradation and crack (joint) behavior response.

In this regard, ethylene vinyl acetate (EVA) synthetic polymer waterproofing sheets are commonly used to meet this performance requirement [2]. EVA performance as a waterproofing membrane has already been academically established to be a suitable material. Holter discusses and compares the sorptivity (water absorption rate) performance of EVA- based membranes for tunnel linings to sprayed cementitious type membranes [3]. Lyapidevskaya discusses the usage of EVA additives in the mixture of a proposed new waterproofing material to increase the elasticity and adhesion performance [4].

However, securing high workability in waterproofing is another question altogether, aside from material performance. Kubal indicates that most waterproofing failures are due to poor workmanship during installation [5], and the lack of standards and guidelines in the installation process and excessive reliance on the experience of the constructors often result in inconsistent results with waterproofing performance, as is the case with EVA waterproofing [6]. In particular, the sheet-based waterproofing work involves the difficult tasks of cutting the waterproofing material on-site, attaching it to a concrete structure and completely bonding the joint between the sheets [7]. Accordingly, the waterproofing success depends on how perfectly the joints between the sheets are bonded together [8]. In this regard, Bucko et al. provides an extensive study on the influence of welding physical conditions on PVC type waterproofing membranes welding quality based on welding speed and welding temperature [9]. According the Bucko et al.’s research, high welding temperatures in general allowed securing high peel strength, but higher welding temperature resulted in the burning of the material.

Further investigation into research trends related to joints were investigated on synthetic polymer sheet waterproofing work and a sheet-membrane composite waterproofing system used for waterproofing in structures. Yilmazer and Park, in different papers, conduct a study on the estimation of joint performance according to jointing methods of sheet membrane waterproofing systems and they confirmed experimentally the physical properties of the joints according to the various methods and materials applied to the sheet [10]. Tomkow et al. study the effect of hydrophobic coatings on the surface of welded structural joints for waterproofing performance [11]. Balkan et al. study the effects of welding methods on PE, PP, and PVC sheets via a hot gas butt welding process [12]. Woo-Il Seo experimentally analyzed the physical properties of the joints of rubber asphalt sheets based on a study on complex waterproofing methods using part of joint watertightness improved sheet [13]. Gun-Woong Go analyzed factors on the optimal joint distance and form of synthetic polymer waterproofing sheets through research on the change of tensile strength with the joint distance and form in the roof composite waterproofing method for synthetic polymer waterproofing sheets [14]. Based on the research results, it was experimentally confirmed that both the joint distance and form affect the tensile performance of the joint. Joo-Hyun Sung [15] insisted that insulation with a concrete surface is more effective at ensuring technical stability than adhesion on the whole surface in the joint treatment method based on an experimental research on a triple composite waterproofing method which applied a loose-laying joint type. Sang-Mook Chang [16] conducted constructability of a waterproofing sheet joint combining an aluminum thin-film and viscosity layer using a high-frequency inductive heating apparatus (2014) and confirmed the constructability of the high-frequency inductive heating apparatus. Jin-Sang Park [17] carried out a fundamental study on induction technology of separation behavior using two-sided adhesion of joints of a composite waterproofing system (2015) and argued that the use of the two-sided adhesion technology in the construction of the waterproofing sheet joint could effectively reduce the problems caused by simultaneous damage to the existing waterproofing layer. Jung-Hoon Lee [18] experimentally analyzed the tensile properties of various reinforcement materials applied to the waterproofing sheet joints of composite waterproofing systems and confirmed the tensile properties of each reinforcement material based on a study on the tensile properties of the reinforcement materials using joints of waterproofing sheet of a composites waterproofing system (2016).

As should be the case, studies reveal that EVA membranes are sensitive to the humidity and ambient temperature condition during installing by hot air welding. Material welding of plastics or similar materials requires a careful process to ensure the quality of the sealing [19]. Badiee et al. shows that while chemical changes do not occur, subtle temperature variation in EVA can cause physical changes and degradation to the EVA properties [20]. When considering common EVA sheet overlap bonding methods, mechanical hot air welding method was preferred in consideration of the quality of stability of the sheet joint, reducing the labor cost and construction period [20]. However, the method is still in the prototypical stage, where due to excessive preference to reduce construction time and labor costs, contractors oftentimes force installation at too high or low ambient temperatures at peak seasons [21]. Due to this factor, leakage defects frequently occur at the joints between the EVA sheets, and this has been a common occurrence in Korea, China and countries in South East Asia [22], where a proper guideline or standard on the installation method of EVA sheets does not yet exist [23]. In theory, EVA hot air welding is considered a dry construction method that requires no separate hardening process to be used, indicating that construction at low or high temperatures is theoretically possible, but optimal welding speed in accordance to the respective ambient temperature conditions construction site must be considered [4]. In this respect, there is not yet sufficient research conducted to provide a database or reference to derive the optimal parameters, thus an experimental study needs to be conducted to quantitatively understand changes in the tensile properties of the EVA sheet joints due to the external temperature and welding speed [18]. Furthermore, review of existing standards on the waterproofing membrane/sheet tensile strength performance (American Society for Testing and Standard and British Standard) shows that these properties are not being tested or evaluated properly in terms of ambient temperature or welding speed [24,25,26,27,28,29,30].

In this regard, this study analyzes and compares the changes to the EVA sheet overlap section tensile strength properties in accordance with different construction temperature conditions (−10, −5, and 0 °C for winter condition, 20 °C for normal conditions, and 25, 30, and 35 °C for the summer condition) and hot air welding speed conditions (3, 4, 5, 6, 7, 8, and 9 m/min). Based on the results, optimal welding speed per temperature condition is outlined whereupon the quality of the joint with a focus on the joint tensile performance can be secured.

### 1.1. Understanding of Synthetic Polymer Sheet Waterproofing Joints

#### 1.1.1. Types of Joining Methods

In general, vulnerable sections in EVA waterproofing work using sheet materials is defects in the joints. Even with sheet materials with excellent demand performance, water leaks occur if the joint stability is not secured, and this in turn makes it impossible to secure the performance of the entire waterproofing layer. Therefore, joint safety in sheet-based waterproofing is very important in the quality control of the overall waterproofing performance.

Table 1 shows the joint construction methods in the waterproofing system using synthetic polymer sheets. These are largely divided into a joint construction method using adhesives, a joint construction method using hot air welding, a joint construction method using tapes, and a joint construction method using fixing nails. This study deals with the joint construction method using hot air welding.

#### 1.1.2. Defects in Sheet Joints

In the case of sheet waterproofing, defects mainly occur at the joints. This is because when exposed to various deterioration factors, the joint is greatly affected by such deterioration factors due to relative difficulties in integration at the joint constructed in the field compared to general parts [21]. Table 2 shows the defects that occur in the joint construction methods.

### 1.2. Effects of Hot Air Welding on Joints

Hot air welding is a method that melts and heats the surfaces of the materials to be joined at a constant temperature and compresses them at the same time to induce integration at the interface of the sheet surfaces [22]. Therefore, it is significantly affected by not only the heating temperature of the heating equipment and time when the material is exposed to temperature (welding speed) but also by the temperature at which construction is carried out (seasonal condition) [23]. These conditions serve as variables, making it difficult to ensure uniform quality [24]. If the proper quality is not achieved at the early stages of construction, the waterproofing layer may lose its waterproofing performance under a wide range of deterioration conditions, such as physical fatigue due to structural behavior during use and temperature behavior due to seasonal changes, and various chemical erosions resulting from the application of underground structures [25]. Therefore, if the joint is formed by applying a uniform method even when hot air welding is used to form the joint, quality deviations will inevitably occur [26]. This deviation in turn serves as another variable and increases the possibility of joint defects as the environmental factors of the use process are added in the long term perspectives. Therefore, this study seeks to analyze the relationship between temperature and hot air welding speed, which can serve as variables in the hot air welding process, and devise a method for securing proper quality.

## 2. Experimental Setup and Tests

### 2.1. Experimental Conditions

In consideration of construction temperatures in winter and summer and at ordinary temperatures, the temperature was set to −10, −5 and 0 °C in winter, 20 °C at an ordinary temperature, and 25, 30 and 35 °C in summer as shown in Table 3. The working speed of the hot air welder was planned to run at speeds ranging from 3 to 9 m/min at 1 m/min intervals for EVA waterproofing sheets exposed to the corresponding temperature for 2 h. In addition, the specimen prepared at the construction temperature and hot air welding speed was settled for at least 24 h at the construction temperature and then set at ordinary temperature for more than 4 h. Refer to Figure 1 for an illustration of the hot air welding method of the EVA sheet.

### 2.2. Performance Evaluation of EVA Sheets Overlap Joints

This study aims to quantitatively determine the relationship between the construction temperature and the hot air welding speed when the joint is formed for EVA sheet waterproofing materials.

For evaluation, test specimens were taken from the specimens with joints formed according to specific temperatures and hot air welding speeds. In order to identify the correlation of the joint tensile performance with the construction temperature and hot air welding speed, tensile performance tests of the membrane joints were carried out. Then, the optimum hot air welding speed and initial joint quality performance were confirmed from the correlation of the joint tensile performance between the two variables.

### 2.3. Specimens Preparation

#### 2.3.1. Waterproofing Materials and Hot Air Welder

Sheets made from EVA are produced and applied in a variety of ways. However, EVA sheets specifically used for the waterproofing method for buildings are applied in a limited manner, particularly in cases of Korea. In this study, evaluation was conducted only on the sheets produced by EVA sheet manufacturers, which are the most actively used in the field of Korea, and the basic properties for these are shown in the following Table 4 below, listing the properties tested and validated by standardized testing methods (KS F 4911 and KS F 4934). As waterproofing materials, sheets incorporating 50% recycled EVA resin were used, and the specifications of these are summarized in Table 4.

Table 5 summarizes the specifications of an automatic hot air welder used for the joint hot air welding process of EVA sheet waterproofing; the pictures of the automatic hot air welder and welding status are shown in Figure 2.

#### 2.3.2. Specimen Preparation

The joint tensile performance test specimens were produced from EVA sheets with joints formed under specific temperatures (−10, −5, 0, 20, 25, 30, and 35 °C) and specific hot air welding speeds (3, 4, 5, 6, 7, 8, 9 m/min). Figure 3 shows a 200 × 50 mm^2^ specimen with a joint length of 50 mm and Figure 4 illustrated the cutting specimen preparation process. The seasonal temperature conditions was set in accordance to the Korean statistics [31], as the experimental parameters were set in accordance to the environmental conditions specific to Korea.

### 2.4. Joint Tensile Performance Test

The tensile performance of the joint should be determined as a value obtained by dividing the force (tensile load) at which the specimen, which is melt-bonded by overlapping two sheet materials, undergoes rupture by the unit area of a part that overlaps. However, it cannot be regarded as the standard area as a difference in the actual melt-bonded area of the overlapping part between specimens occurs in the process of using the hot air welder. In addition, since the method in which the hot air welder melts and bonds the inside of the sheet overlapping part is close to the linear method, it is reasonable to obtain the rupture resistance at the melt-bonded unit length (mm). Therefore, in this study, the tensile performance was determined in units of N/mm.

The tensile performance test was conducted by stretching both ends of the specimen at a speed of 100 mm/min at room temperature (20 ± 2 °C, 65 ± 20%) until the specimen underwent rupture (process is illustrated in Figure 5 below), and it was represented as the average value of three specimens and calculated using Equation (1), as follows.
*T_S_ = P_S_/W_S_*(1)
where: *T_S_*—joint tensile strength (N/mm); *P_S_*—maximum load (N); *W_S_*—specimen width (50 mm)

## 3. Test Results and Discussion

### 3.1. Joint Tensile Strengths According to Temperature

The tensile strength values and seasonal average values of the EVA waterproofing sheet joint specimen using the hot air welder were measured under different construction temperatures and hot air welding speed conditions. The results are summarized in Table 6.

#### 3.1.1. Relationship Between Tensile Strength and Hot Air Welding Speed in Winter

The average value of tensile strengths by season in Table 6 and the change of tensile strengths in winter (−10, −5, and 0 °C) shown in Figure 6 reveal that the tensile performance tends to decrease even further as the ambient temperature decreases, but the welding speed increases.

The specimen at 0 °C shows a maximum of 25.0% reduction in tensile performance compared to the specimen at an ordinary temperature (20 °C, green); the specimen at −5°C shows a 6.1% to 30.9% reduction in tensile performance compared to the specimen at an ordinary temperature (20 °C, green); and the specimen at −10 °C shows a 6.5% to 42.6% reduction in tensile performance compared to the specimen at an ordinary temperature (20 °C, green). Therefore, it is confirmed that the tensile performance of the specimen in winter (−10, −5, 0 °C) decreases by 5.8 to 32.8% on average compared to the specimen at an ordinary temperature (20 °C, green).

The tensile performance in winter decreases faster than that in ordinary temperature conditions at speeds of up to 3~5 m/min, while the performance decreases gradually at speeds of 6 m/min or more.

This is due to a phenomenon that the adhesive force is reduced as the melting heat of the hot air welder delivered to the sheet is cooled in winter. Therefore, it is desirable to avoid construction in winter because the tensile performance of the joint in winter is lower compared to that at ordinary temperatures in all temperature and speed ranges.

#### 3.1.2. Relationship between Tensile Strength and Hot Air Welding Speed in Summer

The average value of tensile strengths by season are shown in Table 6 and the change of tensile strengths in summer (25, 30, and 35 °C) shown in Figure 7 reveal that the tensile performance increases as the ambient temperature rises, and the welding speed increases.

The specimen at 25 °C shows a 2.2 to 20.9% increase in tensile performance compared to the specimen at the ordinary temperature (20 °C, green), the specimen at 30 °C shows a 2.4 to 23.5% increase in tensile performance compared to the specimen at the ordinary temperature (20 °C, green), and the specimen at 35 °C shows a 2.2% to 22.6% increase in tensile performance compared to the specimen at an ordinary temperature (20 °C, green). Therefore, it is confirmed that the tensile performance of the specimen in summer (25, 30, and 35 °C) increases by 3.2 to 22.3% on average compared to the specimen at an ordinary temperature (20 °C, green).

The results according to the welding speed reveal that the tensile performance increases at a speed of up to 7 m/min, but it decreases gradually at speeds of 8m/min or more. This is due to a phenomenon that the adhesive force increases as the heat of the hot air welder delivered to the sheet rises due to the effects of the high ambient temperatures in summer. Therefore, although the joint exhibits high tensile strengths in all temperature ranges, it is desirable to avoid construction at speeds of 8 m/min or more in summer.

## 4. Optimum Hot Air Welding Speed for Each Construction Temperature

Based on the joint tensile performance test results in winter and summer, this study seeks to propose a method for estimating the optimum hot air welding speed for securing the stability of the joints at construction sites where temperature changes occur in various ways.

The welding speed was presented as a speed range that can secure the range of up to 90% based on the highest joint tensile strength for each temperature condition after a speed estimation equation was deduced from regression analysis for each temperature condition on the basis of the above test results. In order to secure more than 80% reliability, the regression equation analysis was conducted on speed ranges of 3 m/min~9 m/min for winter conditions and on speed ranges of 5 m/min~9 m/min at which the highest joint tensile strength was achieved for summer conditions.

### 4.1. Optimum Hot Air Welding Speed According to Construction Temperature in Winter

Table 7 shows the regression equations for each temperature condition deduced from the results of the regression analysis on the experimental results (Table 6) of construction temperature conditions in winter (−10, −5, and 0 °C). The hot air welding speeds for each temperature condition were analyzed based on the deduced regression equations. The analysis found that the optimum hot air welding speed is 3~4.1 m/min at −10 °C, 3~4.6 m/min at −5 °C, and 3~4.9 m/min at 0 °C, as shown in Table 8. In addition, Figure 8 was derived based on the regression equations in Table 7 and Table 8. However, it is advised not to proceed with construction in winter when lower strengths than the tensile strengths of joints at ordinary temperatures are observed in all temperature and speed ranges. However, in the case of inevitable construction, it is necessary to comply with the optimum hot air welding speed range for each temperature condition as described above.

### 4.2. Optimum Hot Air Welding Speed According to Construction Temperature in Summer

Table 9 shows the regression equations for each temperature condition deduced from the results of the regression analysis on the experimental results (Table 3) of construction temperature conditions in summer (25, 30, and 35 °C). The hot air welding speeds for each temperature condition were analyzed based on the deduced regression equations. The analysis found that the optimum hot air welding speed is 4.3~9.0 m/min at 25 °C, 4.7~8.7 m/min at 30 °C, and 5.2~8.6 m/min at 35 °C in summer as shown in Table 10. In addition, Figure 9 is derived based on the regression equations in Table 9 and Table 10.

### 4.3. Comparative Analysis of Winter and Summer Seasonal Temperature

Figure 10 shows the results of tensile strengths according to changes in hot air welding speeds (Table 6) for each temperature condition of the EVA joint specimen. Looking at the joint tensile strengths in winter and summer based on an ordinary temperature (20 °C, green), it can be confirmed that the tensile performance of the specimen in summer (25, 30, and 35 °C) is higher than that of the specimen at an ordinary temperature (20 °C, green), whereas the tensile performance of the specimen in winter (0, −5, and −10 °C) is lower than that of the specimen at an ordinary temperature (20 °C, green).

Based on the results, a comparative analysis on the overall overlap joint tensile strength changes during the winter and summer season can be derived relative to the normal temperature condition (20 °C) tensile strength as an averaged ratio. As can be seen in the results in Table 11, tensile strength will increase during the summer season and decrease during the winter season. During the winter season conditions however, the decrease in the tensile strength is substantially higher from the welding speed of 5 m/min, decreasing from 25~30% in tensile strength, whereas the tensile strength is similar with welding speeds from 3~4 m/min (decrease of only 6~8%). Summer season conditions show an interspersed increase of tensile strength throughout the welding speed conditions, generally ranging from as low as 3% up to 22% on average. Refer to Table 11 for details.

The above results are also expressed in a 3D graph that shows a comprehensive correlation between the EVA sheet joint overlap tensile strength changes to temperature condition and welding speed. As can be seen in Figure 11a, there is a general consistency of tensile strength throughout the welding speed and temperature conditions (with occasional peaks and increases of up to 22~23% in tensile strength). This is indicative of the factor that when EVA sheets are installed in temperate regions similar to Korea, construction during the summer should ensure high strength of EVA sheet roofing. However, the results clearly indicate that installation during winter is not recommended in lower temperature conditions. If in case winter installation cannot be avoided, Figure 11b shows that a low welding speed is recommended, as the tensile strength is similar to the ranges found during normal temperature conditions. Refer to Figure 11 below for details.

## 5. Conclusions

The present research is a basic study carried out to establish an appropriate construction method for the joint on EVA waterproofing sheets. Trends in changes in the quality of joints were investigated by applying not only the variety of construction temperatures and hot air welding speed conditions but also complex deterioration conditions, focusing on the tensile performance of the joints. Based on this, research was conducted for the purpose of presenting the most appropriate joint construction conditions to ensure proper quality. The results showed that quality deviations occur depending on the temperature at which the construction is done during the hot air welding process of the EVA waterproofing sheet joint. This is because the heated air discharged from the equipment causes heat loss according to the temperature. In particular, it was found that low temperatures in winter greatly increase heat loss, resulting in a relatively large decrease in tensile performance compared to performances in summer.

In addition, it was confirmed that the hot air welding speed serves as one of the factors that causes quality deviation because the amount of heat transferred to the interface between the materials varies depending on the welding speed. The quality deviation was more prominent in the specimens prepared under construction conditions in winter than those in summer, and the tensile strength of the specimen in winter further decreased as the hot air welding speed increased. Based on the research results, the optimum hot air welding speed ranges to ensure proper quality according to the construction temperature of the EVA waterproofing sheet are presented as follows. In winter, the optimum speed range is 3~4.1 m/min at −10 °C, 3~4.6 m/min at −5 °C, and 3~4.9 m/min at 0 °C, whereas in summer, the optimum speed range is 4.3~9.0 m/min at 25 °C, 4.7~8.7 m/min at 30 °C, and 5.2~8.6 m/min at 35 °C. However, it is advised not to proceed with construction in winter when lower strengths than the tensile strengths of joints at ordinary temperatures are observed in all temperature and speed ranges. However, in the case of inevitable construction, it is required to comply with the optimum hot air welding speed range for each temperature condition as described above. This study has its significance in that it presents the approximate ranges of construction conditions (welding speed according to temperature) for ensuring the stable quality of the joints of EVA waterproofing sheets, which have actively been applied as waterproofing materials in recent years due to the easy utilization of recycled materials. Nevertheless, the experimental results in this study were able to successfully demonstrate that ambient temperature, both during cold and hot seasons, and the welding speed, affect the adhesion performance of the EVA sheet, and this result should be made known to contractors and those performing the construction using EVA waterproofing.

It should still be noted that the research scope was limited to one EVA waterproofing sheet among other existing synthetic polymer sheets, and only the joint tensile strength was considered as an experimental item to investigate changes in tensile performance in Korea. Therefore, in order to improve the reliability of hot air welding in the construction of synthetic polymer sheets, further research needs to be conducted. Issues regarding the seasonal conditions and welding speed affecting the integrity of the EVA waterproofing layer is not limited to Korea alone, and while the data itself may not be viable in other nations, the demonstrated methodology can be adopted in the international setting to clarify the property change mechanisms of EVA waterproofing in accordance to the international requirements.

## Figures and Tables

**Figure 1 materials-13-05586-f001:**
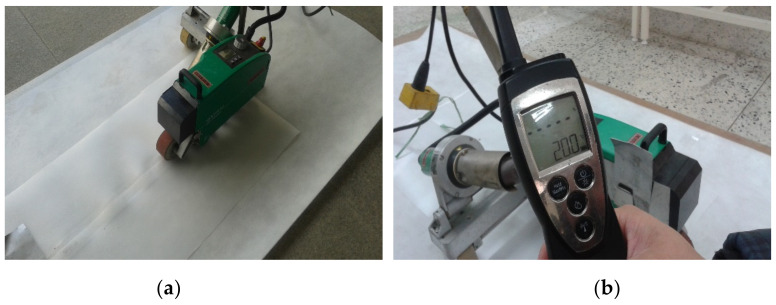
Status of hot air welding construction: (**a**) hot air welding construction; and (**b**) measurement of external temperature.

**Figure 2 materials-13-05586-f002:**
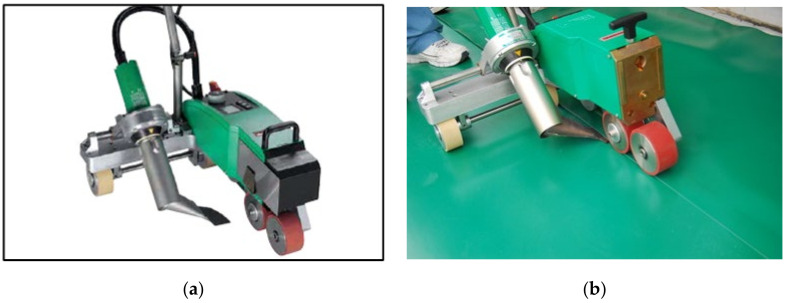
Automatic hot air welder and welding status: (**a**) automatic hot air welder; and (**b**) hot air welding status.

**Figure 3 materials-13-05586-f003:**
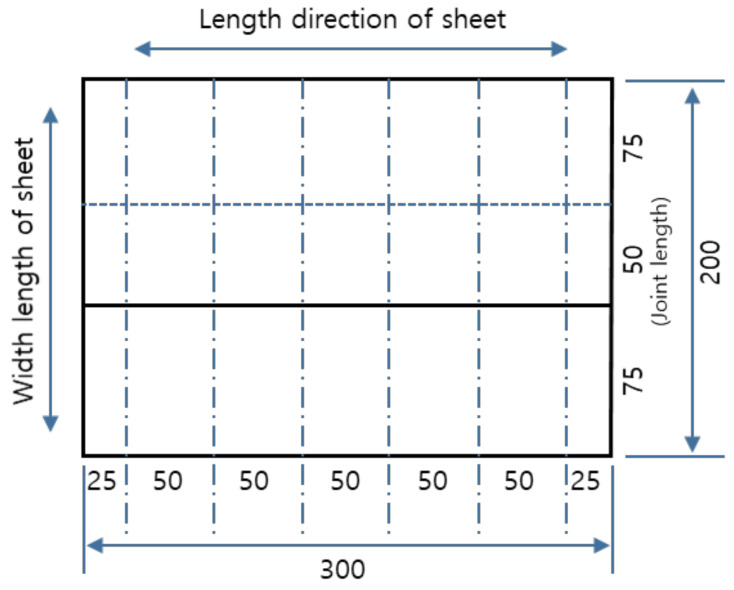
Fabrication of joint tensile performance test specimens (unit: mm).

**Figure 4 materials-13-05586-f004:**
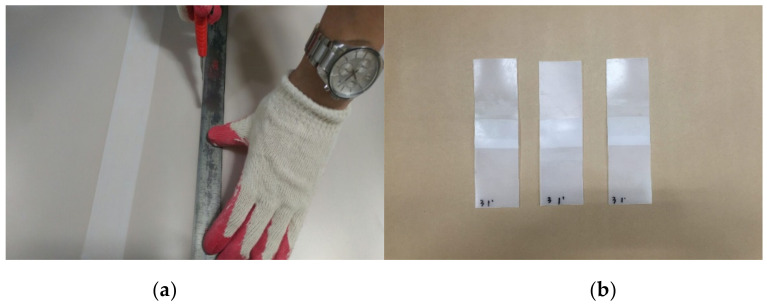
Joint hot air welding construction status: (**a**) specimen fabrication status; and (**b**) a joint tensile performance test specimen.

**Figure 5 materials-13-05586-f005:**
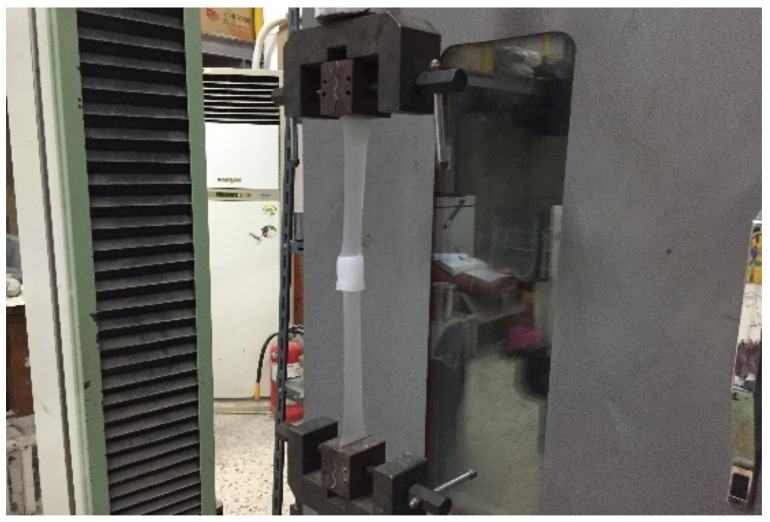
Joint tensile performance test status.

**Figure 6 materials-13-05586-f006:**
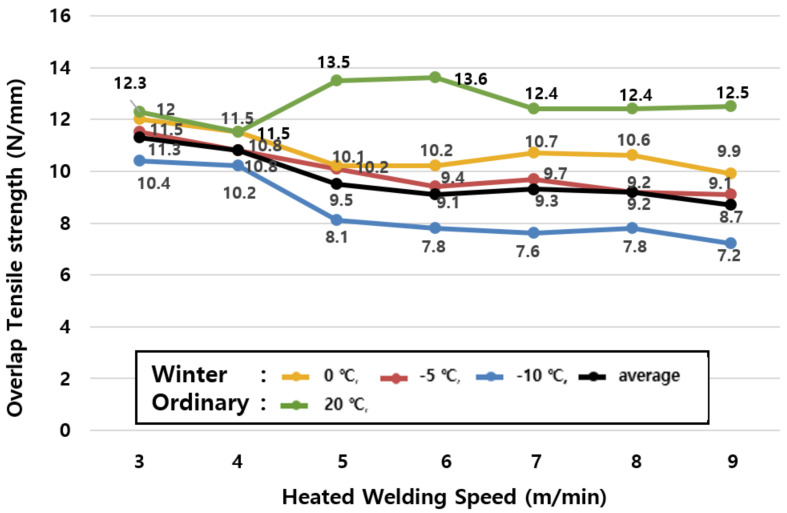
Tensile strength test results by joint construction temperature (winter season).

**Figure 7 materials-13-05586-f007:**
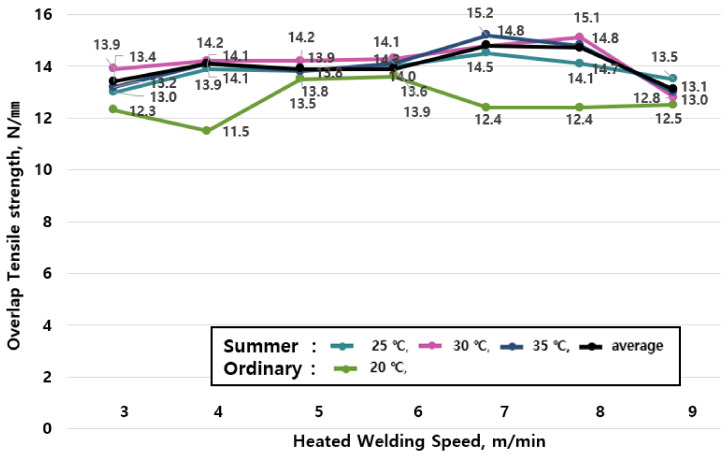
Tensile strength test results by joint construction temperature (summer season).

**Figure 8 materials-13-05586-f008:**
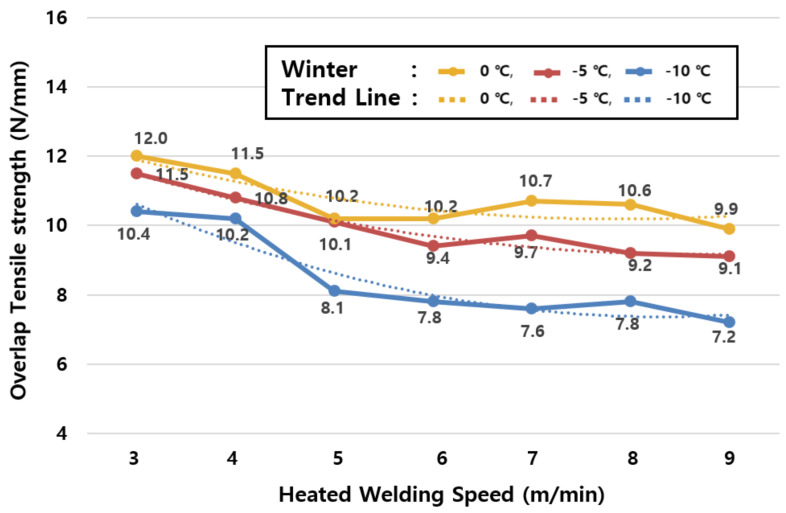
Analysis of optimum hot air welding speed by temperature in winter.

**Figure 9 materials-13-05586-f009:**
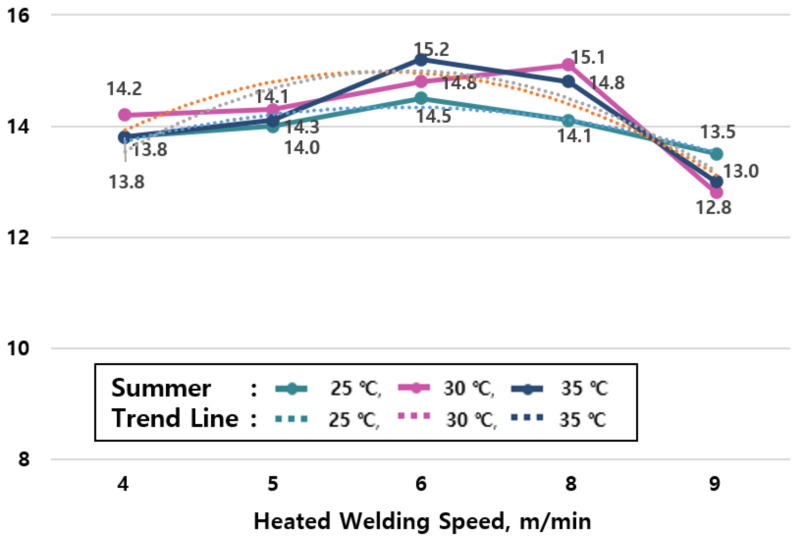
Analysis of the optimum hot air welding speed by temperature in summer.

**Figure 10 materials-13-05586-f010:**
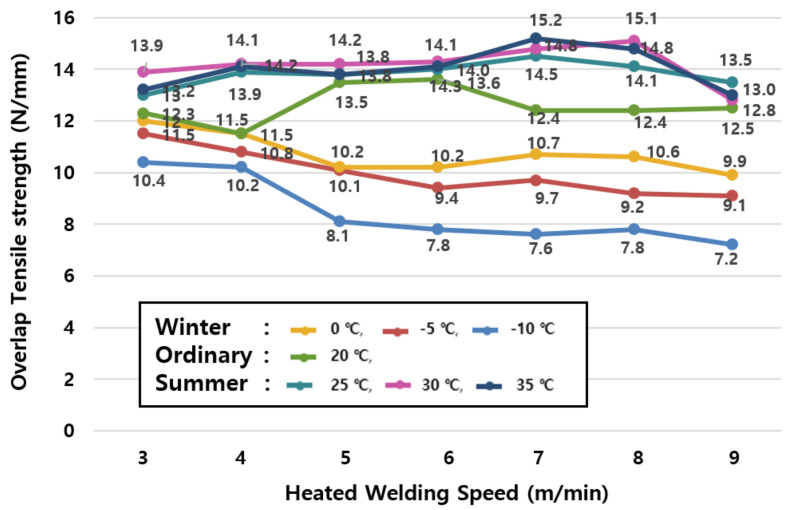
Tensile strength test results according to the joint construction temperature.

**Figure 11 materials-13-05586-f011:**
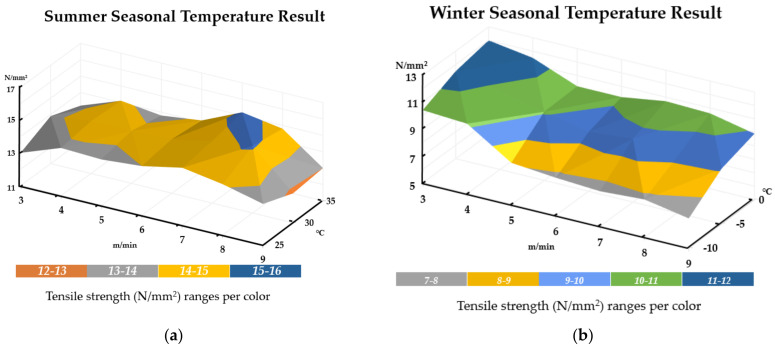
Comparative results showing tensile strength changes relative to welding speed and temperature: (**a**) summer season condition, and (**b**) winter season condition.

**Table 1 materials-13-05586-t001:** Common ethylene vinyl acetate (EVA) overlap joint types.

Title	Illustration	Construction Image	Base Information
Bonding Type	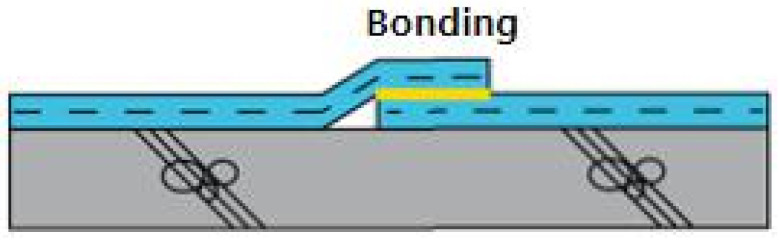	* 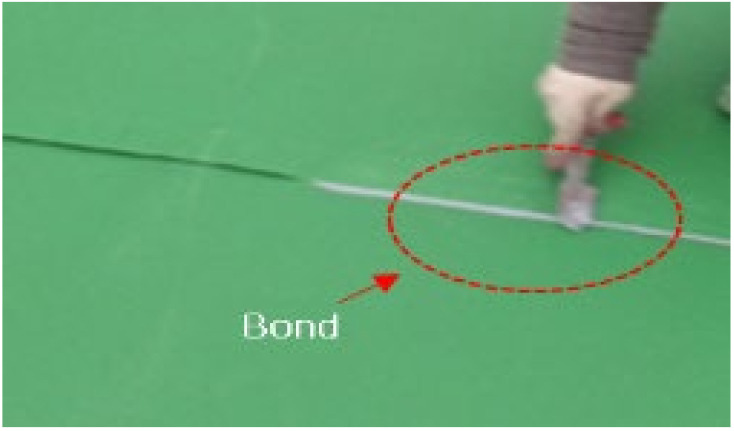 *	Sheet joints are bonded using adhesives
Heated-Air Welding Type	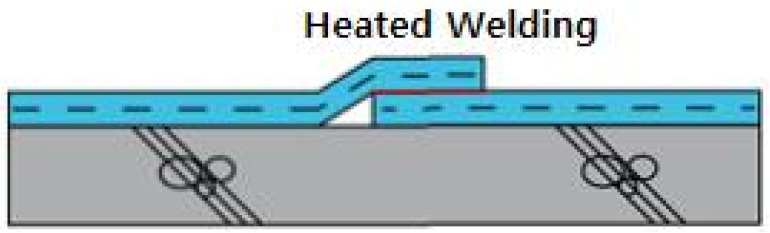	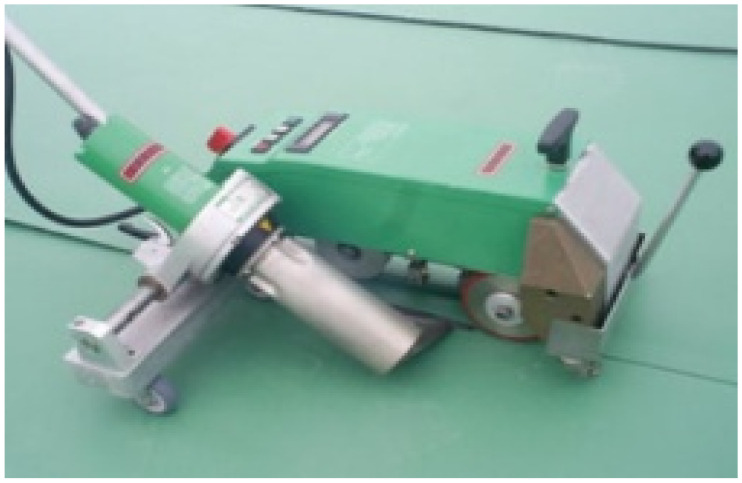	Sheet joints are melted with a hot air welder and bonded together
Taping Type	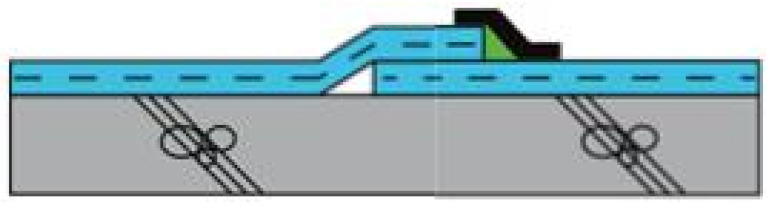	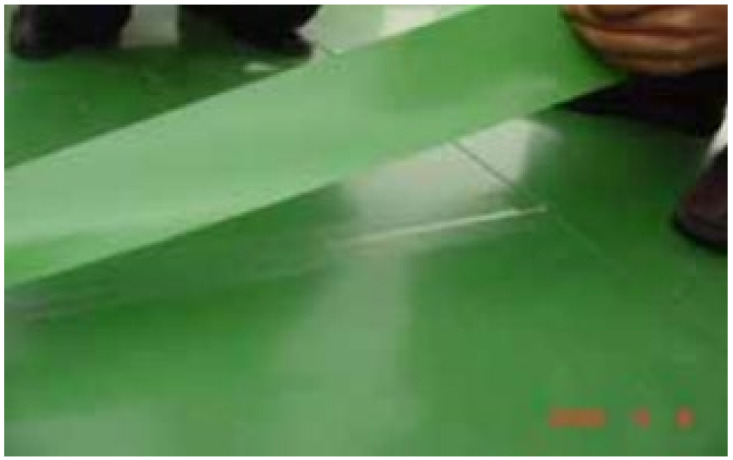	Sheet joints are bonded using tape
Fixing Type	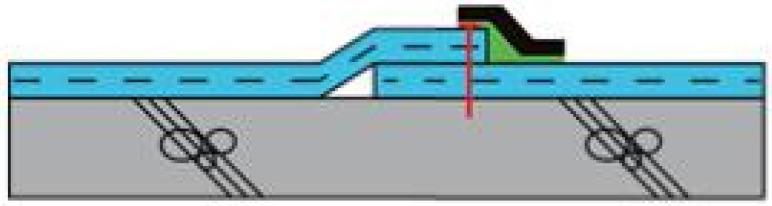	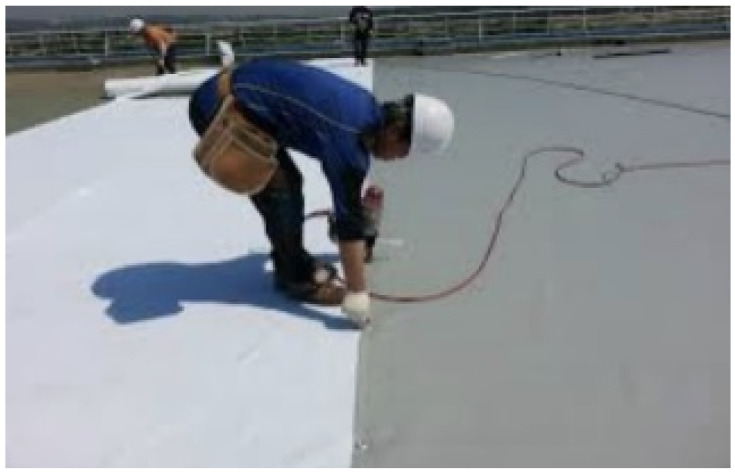	Sheet joints are fixed using fixing nails, and are then taped

**Table 2 materials-13-05586-t002:** Defects in joint construction methods.

Title	Defect Image	Base Information
Bonding Type	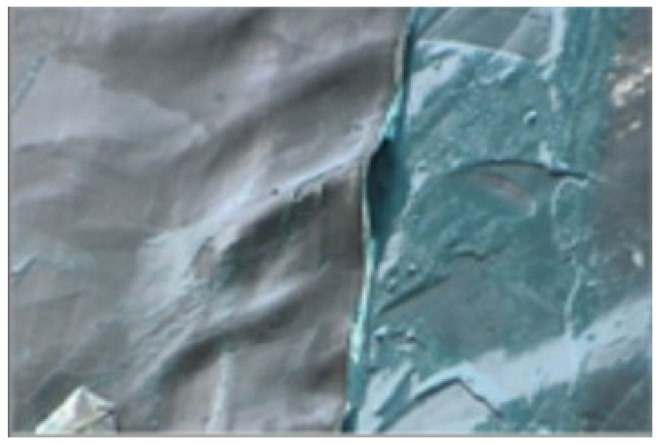	No separate equipment is required, but quality deviations occur depending on the amount of adhesive used and the proficiency levels of workers. Due to different thermal expansion coefficients between the adhesive and the sheet, the joint becomes unfastened and peels off as the temperature changes.
Heated-Air Welding Type	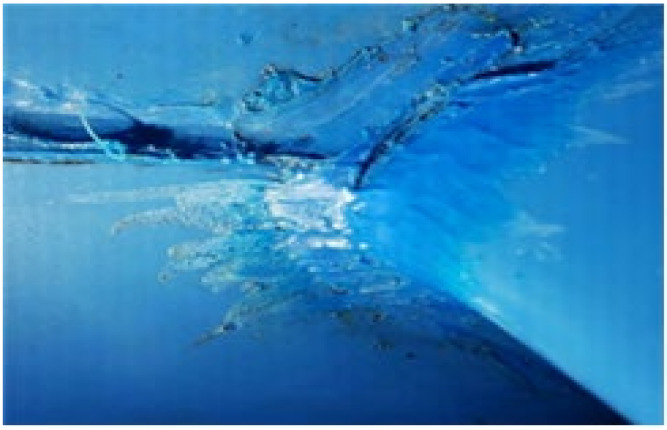	As construction is carried out using equipment that generates heated air, the joint suffers changes in physical properties when exposed to the sheet for a certain period of time. Quality deviations depending on the proficiency levels of workers and may occur in the construction of special areas such as corner parts.
Taping Type	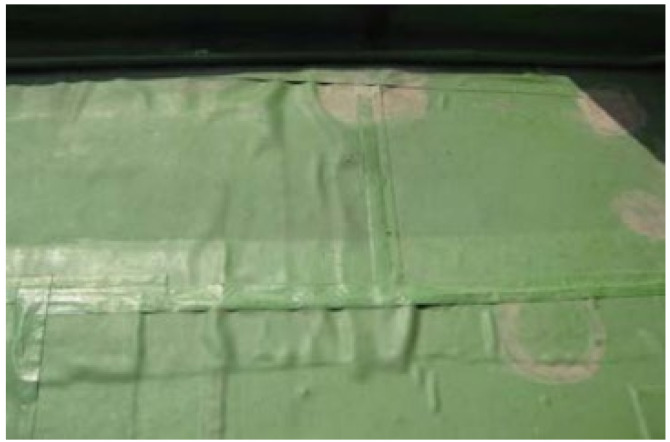	Due to different thermal expansion coefficients between the adhesive tape and the sheet, the joint becomes unfastened and peels off as the temperature changes. It is difficult to secure long-term waterproofing stability as the durability of the joint is dependent on the durability of the tape.
Fixing Type	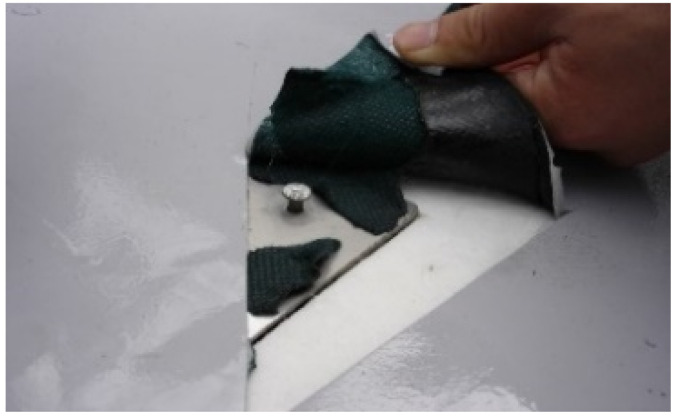	The sheet is damaged by fixing nails as the joint is formed using the fixing nails. The structure undergoes damage as the fixing nails are driven straight into the base surface of the structure, and the damaged part forms a waterway when defects occur in the waterproofing layer.

**Table 3 materials-13-05586-t003:** Temperature and hot air welding speed conditions.

Division	Temperature and Heated Welding Speed						
Temperature (°C)	Winter Condition			Normal Condition	Summer Condition		
	−10,	−5,	0	20	25,	30,	35
Heated Welding Speed	3, 4, 5, 6, 7, 8, and 9 m/min						

**Table 4 materials-13-05586-t004:** EVA waterproofing sheet materials.

Division			Specifications	EVA Sheet
Basic Property	Tensile Strength (N/mm2)	Length	25	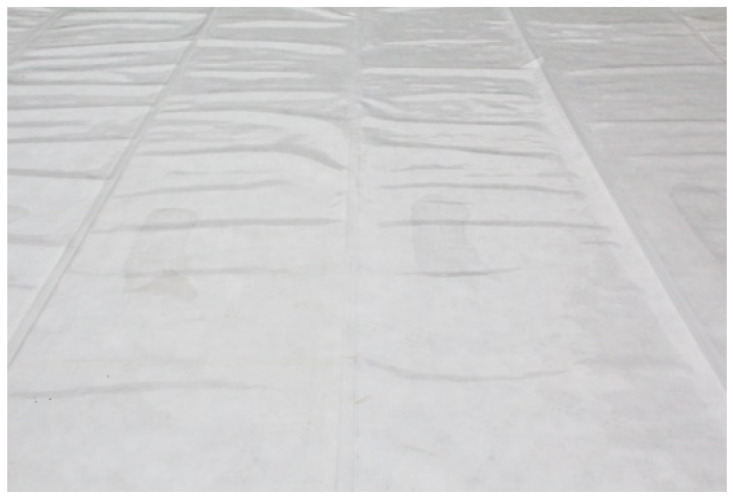
		Width	23	
	Elongation (%)	Length	703	
		Width	707	
	Tear Resistance (N/mm)	Length	76	
		Width	79	
Melting Point			75 °C	
Boiling Point			200 °C	
Specific Gravity			0.948 g/mL at 25 °C	
Flash Point			260 °C	
Components	New EVA		50%	
	Recycled EVA		50%	
Sheet Type			Thermoplastic	
Thickness			1.2 mm	

**Table 5 materials-13-05586-t005:** Material information for fabrication of test specimens.

Specification Items	Unit	Specifications
Voltage	V	230/400
Power	W	3680/5700
Temperature	°C	100~600
Speed	m/min	0.7~12
Air flow range	%	50~100
Width of welding nozzle	mm	40

**Table 6 materials-13-05586-t006:** Joint tensile strengths under different hot air welding speeds and temperature conditions.

Division	Temperature (°C)	Heated Welding Speed (m/min)							Related Standard
		Overlap Tensile Strength (N/mm)							
		3	4	5	6	7	8	9	
Winter Season	−10	10.4	10.2	8.1	7.8	7.6	7.8	7.2	KS F 4917
	−5	11.5	10.8	10.1	9.4	9.7	9.2	9.1	
	0	12.0	11.5	10.2	10.2	10.7	10.6	9.9	
	average	11.3	10.8	9.5	9.1	9.3	9.2	8.7	
Ordinary Section	20	12.3	11.5	13.5	13.6	12.4	12.4	12.5	
Summer Season	25	13.0	13.9	13.8	14.0	14.5	14.1	13.5	
	30	13.9	14.2	14.2	14.3	14.8	15.1	12.8	
	35	13.2	14.1	13.8	14.1	15.2	14.8	13.0	
	average	13.4	14.1	13.9	13.9	14.8	14.7	13.1	

**Table 7 materials-13-05586-t007:** Regression analysis results of hot air welding speed according to the temperature in winter.

Division		Regression Equation	R^2^ = reliability
Winter season	−10 °C	y = 0.1155x^2^ − 1.456x + 11.957	R^2^ = 0.89
	−5 °C	y = 0.0714x^2^ − 0.9571x + 12.371	R^2^ = 0.96
	0 °C	y = 0.0714x^2^ − 0.8429x + 12.671	R^2^ = 0.84

**Table 8 materials-13-05586-t008:** Optimum hot air welding speed range compared to joint tensile strength in winter.

Division			Optimum Hot Air Welding Speed Range Compared to Tensile Strength
Winter season	−10 °C	Tensile strength (N/mm)	(90% of maximum value) 9.4 < y < 10.4 (maximum value)
		Welding speed (m/min)	3 < x < 4.1
	−5 °C	Tensile strength (N/mm)	(90% of maximum value) 10.4 < y < 11.5 (maximum value)
		Welding speed (m/min)	3 < x < 4.6
	0 °C	Tensile strength (N/mm)	(90% of maximum value) 10.8 < y < 12.0 (maximum value)
		Welding speed (m/min)	3 < x < 4.9

**Table 9 materials-13-05586-t009:** Regression analysis results of hot air welding speed according to temperatures in summer.

Division		Regression Analysis	R^2^ = Reliability
Summer season	25 °C	y = −0.1786x^2^ + 1.0214x + 12.88	R^2^ = 0.86
	30 °C	y = −0.3571x^2^ + 1.9429x + 12.34	R^2^ = 0.80
	35 °C	y = −0.4071x^2^ + 2.3529x + 11.6	R^2^ = 0.81

**Table 10 materials-13-05586-t010:** Optimum hot air welding speed range compared to joint tensile strength in summer.

Division			Optimum Hot Air Welding Speed Range Compared to Tensile Strength
Summer season	25 °C	Tensile strength (N/mm)	(90% of maximum value) 13.1 < y < 14.5 (maximum value)
		Welding speed (m/min)	4.3 < x < 9.0
	30 °C	Tensile strength (N/mm)	(90% of maximum value) 13.6 < y < 15.1 (maximum value)
		Welding speed (m/min)	4.7 < x < 8.7
	35 °C	Tensile strength (N/mm)	(90% of maximum value) 13.7 < y < 15.2 (maximum value)
		Welding speed (m/min)	5.2 < x < 8.6

**Table 11 materials-13-05586-t011:** Tensile strength change compared to the normal temperature condition (20 °C).

Division	Temperature (°C)	Tensile Strength Change Ratio in Accordance to Welding Speeds (%)						
		3	4	5	6	7	8	9
Winter Season	−10	85	89	60	57	61	63	58
	−5	93	94	75	69	78	74	73
	0	98	100	76	75	86	85	79
	Average Tensile Strength Change	−8	−6	−30	−33	−25	−26	−30
Summer Season	25	106	121	102	103	117	114	108
	30	113	123	105	105	119	122	102
	35	107	123	102	104	123	119	104
	Average Tensile Strength Change	9	22	3	4	20	18	5

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
