# Peer review of "Analysis on the Effects of External Temperature and Welding Speed on the Safety of EVA Waterproofing Sheet Joints by Hot Air Welding"

_materials, 2020, doi:10.3390/ma13235586_

Round 1

Reviewer 1 Report

The contribution is a technical paper where most of the work is only descriptive in nature. However, the contribution should be published in “materials” after minor revision because the paper gives some new insight into which welding conditions are the most useful to prepare high-strength EVA waterprofing sheet joints by hot air welding.

  • Check the whole paper carefully to insert spaces where necessary.
  • The arrangement between text and/or figures and/or tables must be improved (see page 2, 8, 9, 10 and 11).
  • Table 3 (line 152): “-10°C”, “-5°C” – Not bold.
  • Line 156: “2.2. Performance evaluation of lap joints” – Shift to the next page (page 6).
  • Table 5 (line 173): Delete column “Remarks” (there is no information).
  • Line 178: Shape of the minus sign must be the same within the whole paper.
  • Line 180: Replace “mm” with “mm2”.
  • Table 6 (line 209) and Table 8 (line 282): Check font of “N/mm”.
  • Table 7 (line 281) and Table (line 293). “R2” – “2” superscript.
  • Line 351: “KIM” – Not in capital letters.
  • Line 355, 358, 372 and 378: “KOREA” – Not in capital letters.
  • The temperatures used in the present contribution are clearly related to Korean climatic conditions (cold subarctic winters and tropical warm-humid summers). In other regions such as in central Europe pleasant temperatures ranging from 10 °C to 20 °C are highly typical for most of the time. Therefore, the results of the paper are somewhat difficult to transfer to other regions.

Author Response

Reviewer 1

The authors of the article would like to extend their utmost appreciation the reviewer for their time and effort spent to improve the quality of the paper.

The authors have prepared the following responses to the comments and points of revision. We have first provided the quote on the specific point specified the reviewer that the authors intend to address, and provided the relevant response. Please refer to the below sections for details;

The contribution is a technical paper where most of the work is only descriptive in nature. However, the contribution should be published in “materials” after minor revision because the paper gives some new insight into which welding conditions are the most useful to prepare high-strength EVA waterprofing sheet joints by hot air welding.

Reviewer Comment 1

Check the whole paper carefully to insert spaces where necessary.

Response 1

Paper has been checked throughout the insert spaces where necessary. Please refer to the revised draft for details.

Reviewer Comment 2

The arrangement between text and/or figures and/or tables must be improved (see page 2, 8, 9, 10 and 11).

Response 2

Figures and tables have been rearranged for better articulation and clarity.

Reviewer Comment 3

Table 3 (line 152): “-10°C”, “-5°C” Not bold.

Response 3

-Bolding removed. Please refer to the revised Table 3 for details.

Reviewer Comment 4

Line 156: “2.2. Performance evaluation of lap joints” Shift to the next page (page 6).

Response 4

-Subtitle 2.2 Performance evaluation of EVA sheets overlap joints is now in line 162 in the revised article.

Reviewer Comment 5

Table 5 (line 173): Delete column “Remarks” (there is no information).

Response 5

Table 5 (line 179 in the revised version), remarks column has been deleted

Reviewer Comment 6

Line 178: Shape of the minus sign must be the same within the whole paper.

Response 6

minus sign changed throughout the paper

Reviewer Comment 7

Line 180: Replace “mm” with “㎟”.

Response 7

The unit for tensile strength is based on linear bonding of the overlap section of the EVA waterproofing membrane sheets, thus the strength determination is not based of surface area (explanation is provided in Lines 197 to 203 in the revised version of the article)

Reviewer comment 8

Table 6 (line 209) and Table 8 (line 282): Check font of “N/㎟”.

Response 8

unit font changed accordingly and Tables 6 and 8

Reviewer comment 9

Table 7 (line 281) and Table (line 293). “R2” “2” superscript.

Response 9

Coefficient labels changed to superscript in Tables 7 and Table 9

Reviewer comment 10

Line 351: “KIM” Not in capital letters.

Line 355, 358, 372 and 378: “KOREA” Not in capital letters.

Response 10

capital letters removed in the references

Reviewer Comment 11

The temperatures used in the present contribution are clearly related to Korean climatic conditions (cold subarctic winters and tropical warm-humid summers). In other regions such as in central Europe pleasant temperatures ranging from 10 °C to 20 °C are highly typical for most of the time. Therefore, the results of the paper are somewhat difficult to transfer to other regions.

Response 11

The introduction (lines 96-100), conditions (subsection 2.3.2, lines 186-188) parts of the result (lines 318 to 319) and conclusion (lines 358 to 366) to stress that the experimental results are reflective of the Korean environmental conditions only. Further explanation has been included in the introduction section (lines 89 to 112) to elaborate that the problems regarding EVA overlap integrity relative to the welding speed and ambient temperature conditions during construction is not limited to Korea alone (the following lines have been included in the paper lines 363-366).

“Issues regarding the seasonal conditions and welding speed affecting the integrity of the EVA waterproofing layer is not limited to Korea alone, and while the data itself may not be viable in other nations, the demonstrated methodology can be adopted in the international setting to clarify the property change mechanisms of EVA waterproofing in accordance to the international requirements.”

The authors would like to express their sincerest gratitude for the reviewers’ efforts and time taken to review and comment on this article. Thanks to your esteemed feedbacks, the authors believe the paper was able to undergo a significant improvement since the previous version. The authors hope that the revised version satisfies the points of inquiries provided by the reviewers

Reviewer 2 Report

Dear sirs,

According to the reviewer, the presented material is useful for technical data of the mannufacturer of the product, and not for scientific article.

Most of the information given in the introduction is generally known and has nothing to do with the scientific annotation of the moving problem. The manuscript even includes photos available on the manufacturer's webside ( in Table 1), without mentioning the source of their origin.

There are no data about the amount of materials tested and their properties.For each measurement point,  the given results are the average of three measurements , which seems to be insufficient population to conclude in this regard.

The reference to the national standard in Table 6 ( i.e. KSF 4917) seems appropriate when the article is to be presented in national publishing house, especially as there are  available standards that unify the research methodology in many countries  ( i.e. EN 12311-2 - shear resistance of joints for plastic flexible sheets for roof waterproofing). The same results are given twice, i.e. in the Table 6 and then in Figires 6, 7 and 8, which do not bring anything new to the discussion.

The entries given in the conclusion , apart from additonal information on the welding speed in comparison with temperature ( which may be useful for manufacturer's technical data sheet) do not bring anything new to the state of knowledge in the field in question known for many years. It is well known that welding of plastic sheets should not be carried out at low temperature, as the effectiveness of this type of work is limited. Such data has been used for many years in technical requirements around the world.

The list of references is very poor, referring only Korean publications, additionally not always matching the subject matter in question ( see item 9 on bituminous products and not plastic sheets).

In conclusion, I propose reject the submitted manuscript.

with kind reagrds

Author Response

Reviewer 2

Dear sirs,

According to the reviewer, the presented material is useful for technical data of the manufacturer of the product, and not for scientific article.

Most of the information given in the introduction is generally known and has nothing to do with the scientific annotation of the moving problem. The manuscript even includes photos available on the manufacturer's webside ( in Table 1), without mentioning the source of their origin.

There are no data about the amount of materials tested and their properties. For each measurement point, the given results are the average of three measurements , which seems to be insufficient population to conclude in this regard.

The reference to the national standard in Table 6 ( i.e. KSF 4917) seems appropriate when the article is to be presented in national publishing house, especially as there are available standards that unify the research methodology in many countries ( i.e. EN 12311-2 - shear resistance of joints for plastic flexible sheets for roof waterproofing). The same results are given twice, i.e. in the Table 6 and then in Figires 6, 7 and 8, which do not bring anything new to the discussion.

The entries given in the conclusion , apart from additonal information on the welding speed in comparison with temperature ( which may be useful for manufacturer's technical data sheet) do not bring anything new to the state of knowledge in the field in question known for many years. It is well known that welding of plastic sheets should not be carried out at low temperature, as the effectiveness of this type of work is limited. Such data has been used for many years in technical requirements around the world.

The list of references is very poor, referring only Korean publications, additionally not always matching the subject matter in question ( see item 9 on bituminous products and not plastic sheets).

In conclusion, I propose reject the submitted manuscript.

with kind regards

Response 2

The authors of the article would like to extend their utmost appreciation the reviewer for their time and effort spent to improve the quality of the paper.

With regards to the reviewer’s opinions, the author would like to express their sincerest apologies and regret for having prepared a lacking article.

We have revised the article such that the scientific soundness and contribution may be more apparent than it was before. If the reviewer would be so kind as to give the article another chance, please refer to the revised introduction and results (Section 4.3) of the article for details of the changes.

More references to the existing international studies have been included, but it must be made clear that there actually aren’t too many studies that discuss the importance of ambient temperature and welding speed for securing high performance EVA sheet overlap section in roofing. part of this issue is because EVA hot air welding is considered a dry construction method that requires no separate hardening process to be used, indicating that construction at low or high temperatures is theoretically possible, but optimal welding speed in accordance to the respective ambient temperature conditions construction site must be considered. In this respect, there is still not yet sufficient research conducted to provide a database or reference to derive the optimal parameters, thus an experimental study needs to be conducted to quantitatively understand changes in the tensile properties of the EVA sheet joints due to the external temperature and welding speed. In said regard, we revised the paper to make the purpose more clearer with more analytical data to support this point.

Thank you kindly for your reconsideration.

Reviewer 3 Report

This paper concentrated on the adhesion safety of joints of EVA waterproofing sheets in building construction. Results showed that the tensile strength of the joint significantly varies depending on the 26 ambient temperature and hot air welding speed. The suitable hot air welding speed was suggested in different temperature conditions.

I think that manuscript is a good paper for application and engineering.
The English writing is not very good, but proper to read and discussion. As for the weak point, I think this manuscript is lack of scientific research, for example the microstructure difference of joints of EVA waterproofing sheets under different conditions in order to search the reason for properties change, and the detailed adhesion properties to describe the change of adhesion safety.

Author Response

  1. Reviewer

Response 3

The authors of the article would like to extend their utmost appreciation the reviewer for their time and effort spent to improve the quality of the paper.

The authors have a new version of the draft, with improvement and clarification on the purpose and scientific soundness of the proposed experimental regime. Please refer to the revised article for details, and if you should have any advice, comments or points of revision that is required, we would be more than happy to reflect them in the next version. Thank you kindly.

Round 2

Reviewer 2 Report

Dear Sirs,

In my opinion , the text submitted for review still needs to be supplemented. There is still no information on the amount of materials used in the tests (whether the obtained results concern one product , from one manufacturer or several different ones). Lack of information on the basic identification properties of the materials/material used in the tests, with particular emphasis on tensile properties , which have a significant impact on the assessment of joint strength.

with kind regarads

Author Response

Dear Reviewer,

Thank you kindly for your efforts and time to review the article once again, and your words of advice and comments are most appreciated. With regards to your comment, we have prepared the following response;

Comment 1

There is still no information on the amount of materials used in the tests (whether the obtained results concern one product , from one manufacturer or several different ones). Lack of information on the basic identification properties of the materials/material used in the tests, with particular emphasis on tensile properties , which have a significant impact on the assessment of joint strength.

Response 2

As there are more than one point of requested revision made in your single comment, we will try to address them as much as possible.

With regards to the information on the basic properties of the EVA sheet, a new table has been included (please refer to the revised version, Line 189 Table 4 for details), wherein tensile strength and tear resistance properties have been included to provide more insight on the material properties. Please keep in mind that all tested materials are compliant to national testing standards (KS F 4911, and KS F 4934, which are adopted from ASTM and JIS standards to begin with). This information has also been included in Lines 181 to 186 in the revised article.

With regards to the choice to materials for the testing, only one product (dominant) of EVA sheet was selected, and the reasoning as to why has been explained again in Lines 181 and 186. The purpose of the article is not just promote this particular EVA sheet, but to shed disclosure about the importance of considering the ambient temperature and welding speed during installation of sheets. In this regard, if you refer to Bucko's paper (https://www.researchgate.net/publication/232729792_Influence_of_welding_physical_conditions_on_waterproofing_membrane_weld_quality), they too only examine one type, or product, of PVC sheet for their experimental regime, as their focus is more around welding temperature, rather than ambient condition. While the similarities are evident, welding temperature is easier to follow, as manufacturer specifications already provide recommended information (nevertheless, Bucko provides a significant contribution), and basic knowledge about asphalt sheets, pvc sheets or EVA sheets is more than enough for experienced workers to be able to secure proper installation. But this is only during temperate conditions.

Sheets during installation period will be inevitably exposed to the outside temperature conditions for extended period of time. While in the U.S. or European nations, avoiding installation of waterproofing during cold periods maybe common sense (there are still several cases where waterproofing injection repair and installations were conducted in below room temperature conditions in the western region), this is particularly not the case for Asian, South East Asian countries, and Middle Eastern countries. Construction period efficiency is prioritized over securing higher durability/quality waterproofing all the time, prompting work to continue even during disadvantageous periods. Which is why, this type of research must be disclosed, even if the scope is limited to only EVA sheets at this point in time.

However pointing this situation out directly in an academic paper is inappropriate, but a brief explanation has been supplemented on this matter in the conclusion section, lines 371 to 383 in the revised article.

We hope that the above response satisfy the requirements of your comments and points of revision.

We would like to express our sincerest gratitude once again for your time.

Thank you